

# Prognostic and chemotherapeutic implications of a novel four-gene pyroptosis model in head and neck squamous cell carcinoma

Peiyang Yuan[*], Sixin Jiang[*], Qiuhao Wang, Yuqi Wu, Yuchen Jiang, Hao Xu, Lu Jiang and Xiaobo Luo

State Key Laboratory of Oral Diseases & National Center for Stomatology & National Clinical Research Center for Oral Diseases & Department of Oral Medicine, West China Hospital of Stomatology, Sichuan University, Chengdu, China

[*] These authors contributed equally to this work.

Corresponding authors
Lu Jiang, jianglu@scu.edu.cn
Xiaobo Luo, luoxbscu@163.com

## ABSTRACT

**Background**. Head and neck squamous cell carcinoma (HNSCC) is one of the most common cancers. Chemotherapy remains one dominant therapeutic strategy, while a substantial proportion of patients may develop chemotherapeutic resistance; therefore, it is particularly significant to identify the patients who could achieve maximum benefits from chemotherapy. Presently, four pyroptosis genes are reported to correlate with the chemotherapeutic response or prognosis of HNSCC, while no study has assessed the combinatorial predicting efficacy of these four genes. Hence, this study aims to evaluate the predictive value of a multi-gene pyroptosis model regarding the prognosis and chemotherapeutic responsiveness in HNSCC.

**Methods**. By utilizing RNA-sequencing data from The Cancer Genome Atlas database and the Gene Expression Omnibus database, the pyroptosis-related gene score (PRGscore) was computed for each HNSCC sample by performing a Gene Set Variation Analysis (GSVA) based on four genes (Caspase-1, Caspase-3, Gasdermin D, Gasdermin E). The prognostic significance of the PRGscore was assessed through Cox regression and Kaplan–Meier survival analyses. Additionally, chemotherapy sensitivity stratified by high and low PRGscore was examined to determine the potential association between pyroptosis activity and chemosensitivity. Furthermore, chemotherapy sensitivity assays were conducted in HNSCC cell lines *in vitro*.

**Results**. As a result, our study successfully formulated a PRGscore reflective of pyroptotic activity in HNSCC. Higher PRGscore correlates with worse prognosis. However, patients with higher PRGscore were remarkably more responsive to chemotherapy. In agreement, chemotherapy sensitivity tests on HNSCC cell lines indicated a positive association between overall pyroptosis levels and chemosensitivity to cisplatin and 5-fluorouracil; in addition, patients with higher PRGscore may benefit from the immunotherapy. Overall, our study suggests that HNSCC patients with higher PRGscore, though may have a less favorable prognosis, chemotherapy and immunotherapy may exhibit better benefits in this population.

## INTRODUCTION

Head and neck squamous cell carcinoma (HNSCC) ranks as the sixth most common cancer globally, with an estimated 830,000 diagnoses and 450,000 deaths reported in 2018 (*Bray et al., 2018*; *Ferlay et al., 2019*; *Johnson et al., 2020*). The primary therapeutic strategies for HNSCC include surgery, chemotherapy, and radiotherapy, with cisplatin and 5-fluorouracil being the first-line chemotherapeutic agents. However, a substantial proportion of HNSCC patients may develop chemotherapeutic resistance, leading to poor clinical outcomes. Given the toxicity induced by chemotherapy, identifying patient populations who are more likely to benefit from this treatment modality is of significant importance.

Pyroptosis, an immunogenic form of cell death characterized by pore formation in the plasma membrane, cell swelling and membrane rupture, results in massive leakage of cytosolic contents and the secondary inflammatory response (*Wang et al., 2020*). Pyroptosis can be induced by the canonical caspase-1 inflammasome or by the activation of caspases-4, -5, and -11, mediated by cytoplasmic lipopolysaccharide 1, 2, and 3 (*Wang et al., 2017*; *Jorgensen & Miao, 2015*), which is followed by the cleavage of gasdermin proteins (Gasdermin D (GSDMD), Gasdermin E (GSDME), and Gasdermin B (GSDMB)) at their middle linkers to release the gasdermin-N terminal, which harbors pore-forming activity (*Ding et al., 2016*). Pyroptosis has been implicated in the therapeutic response of tumors to chemotherapy (*Inoue & Tani, 2014*; *Aoto et al., 2018*; *Zhang et al., 2021*; *Gao et al., 2020*) and immunotherapy (*Zhou et al., 2020*; *Gao et al., 2022*; *Zhang et al., 2022*). Several previous studies have evaluated the predictive or prognostic value of pyroptosis in HNSCC by employing least absolute shrinkage and selection operator (LASSO) Cox regression analyses based on various pyroptosis-related gene or lncRNA signatures (*Lu et al., 2022*; *Shen et al., 2021*; *Zhu et al., 2021*; *Li et al., 2022*); however, the studies screened out genes that were more highly correlated with patients' survival, but ignoring the direct representing potential of these genes for pyroptotic activity. So far, four pyroptosis-related genes are reported to implicate in the progression of HNSCC. Initially, it has been suggested that tumor cells may undergo pyroptosis induced by chemotherapy in a caspase-3-dependent manner, and GSDME-mediated pyroptosis has been correlated with the response of oral cancer to chemotherapy (*Wang et al., 2022*; *Rioja-Blanco et al., 2022*; *Zi et al., 2023*); GSDMD and caspase-1 are also reported to implicate in mediating pyroptosis which regulates the progression of HNSCC (*Zhang et al., 2020*; *Wang & Liu, 2023*; *Yue et al., 2019*). Therefore, we are motivated to construct a pooled pyroptosis score to reflect the actual pyroptotic level, which is still a blank in the field.

Thus, this study aims to fill this gap by constructing a four pyroptosis-related gene model called the pyroptosis-related gene score (PRGscore), reflecting overall pyroptosis activity, to predict HNSCC prognosis and therapeutic sensitivity to chemotherapy and immunotherapy.

## MATERIALS AND METHODS

### Data extraction and analysis

RNA-sequencing data (497 tumor samples and 44 normal samples, Transcripts Per Million value) were extracted from The Cancer Genome Atlas (TCGA) database (https://portal.gdc.cancer.gov/), specifically focusing on the expression levels of four PRG genes (CASP1, CASP3, GSDME, GSDMD). Further validation analysis was carried out using the GSE41613 cohort, which includes 97 HNSCC patients, deriving from the Gene Expression Omnibus (GEO). Any batch effect resulting from nonbiotech bias was corrected using the "combat" algorithm from the "sva" R package. Corresponding clinicopathologic data (including clinical stage, age, sex, grade, T-stage, N-stage, overall survival (OS) time, and status) were also collected.

### Generation of pyroptosis-related gene score (PRGscore)

Four pyroptosis-related genes (CASP1, CASP3, GSDME, GSDMD), previously indicated to be involved in the initiation and development of HNSCC, were defined as the pyroptosis-related gene set (*Wang et al., 2022*; *Rioja-Blanco et al., 2022*; *Zi et al., 2023*; *Zhang et al., 2020*; *Wang & Liu, 2023*; *Yue et al., 2019*). A human gene transfer format (GTF) file was obtained from the Ensembl dataset (http://asia.ensembl.org) for gene ID annotation, which enabled the identification of specific mRNAs.

The expression levels of these pyroptosis-related genes were extracted and compared between HNSCC and corresponding normal tissue using the "limma" package. A gene set variation analysis (GSVA) index for each sample was generated by calculating a gene set enrichment score using the z-score augmenting method. The GSVA index was then considered as the PRGscore, representing each sample's pyroptotic level. Pearson correlation analysis was conducted to compute the correlation between gene expression levels and the GSVA index. An association was considered significant if the correlation coefficient was >0.1 and $P$ was < 0.05.

### Prognostic analysis based on the PRGscore

The "surv_cutpoint" function from the "survminer" R package was used to determine the ideal cut-off values. These values were used to categorize samples into high or low PRGscore groups. Associations between the PRGscore and clinicopathological parameters were assessed using the chi-squared test (Table S1). A risk curve was plotted using R software to display the distinct survival status associated with high or low PRGscore for individuals from the TCGA or GEO cohorts. Kaplan–Meier survival and multivariable Cox proportional hazard regression analyses were performed to evaluate the PRGscore's predictive value. To verify the COX proportional hazards assumption, the global proportional hazards assumption test using "cox.zph()" function, and graphed scaled Schoenfeld residuals of each covariate against the transformed time using "ggcoxzph()" function were applied. The prognostic value of the PRGscore was further validated using multivariable Cox regression and Kaplan–Meier survival analyses in the GSE41613 dataset.

## IC50 estimation of routine chemotherapeutic drugs for HNSCC

The half-maximum inhibitory concentration (IC50) of six commonly used chemotherapeutic drugs for HNSCC (cisplatin, methotrexate, gemcitabine, 5-fluorouracil, paclitaxel, and Docetaxel) was estimated using the "pRRophetic" package (*Geeleher, Cox & Huang, 2014*). The "pRRophetic" R package is typically used to predict clinical chemotherapeutic responses using gene expression and drug sensitivity data from cell lines in the Cancer Genome Project (CGP). After extracting data from the CGP, a ridge regression model was established using certain genes as predictors and drug sensitivity (IC50) values as the outcome variable. Subsequently, the processed, standardized, filtered clinical tumor expression data (HNSCC data obtained from TCGA) was employed in this model to estimate drug sensitivity.

## Immunotherapeutic efficacy predicting analysis

The study involved an analysis of the varying expression levels of immune checkpoint inhibitor -related genes, specifically Programmed Cell Death 1 Ligand 1(CD274), Cytotoxic T-Lymphocyte-Associated Protein 4(CTLA4), Programmed Cell Death 1(PDCD1), Indoleamine 2,3-Dioxygenase 1(IDO1), Programmed Cell Death 1 Ligand 2(PDCD1LG2), and Hepatitis A Virus Cellular Receptor 2(HAVCR2) between two cohorts. Additionally, to gauge the potential response to immunotherapy, the Immunopheno score (IPS) score from the TCGA-HNSC project was acquired through The Cancer Immunome Atlas (TCIA) database (https://tcia.at/home). This scoring was computed based on the gene expression data of immune-related genes.

## Cell culture

Fourteen HNSCC cell lines, namely HSC-4, CAL-27, UM1, HN4, HN12, HN30, CAL-33, SCC-47, SCC-9, H314, H103, H357, H376, and SCC-25 were utilized for this study. Eight of these lines (HSC-4, CAL-27, UM1, HN4, HN12, HN30, CAL-33, and SCC-47) were maintained in DMEM/high-glucose (Cytiva, Logan, USA) supplemented with 10% fetal bovine serum (FBS, VivaCell, Shanghai, China) and 1% streptomycin–penicillin (Biosharp, Hefei, China). The remaining six lines, SCC-9, H314, H103, H357, H376, and SCC25, were grown in F12 medium (VivaCell, Shanghai, China) enriched with 10% FBS, 1% streptomycin–penicillin, and 0.5 μg/mL hydrocortisone. The cells were incubated in a humidified environment at 37 °C with 5% CO2. All these cells were carefully maintained in the central cell bank of State Key Laboratory of Oral Diseases, West China Hospital of Stomatology in Sichuan University. Cell lines were routinely tested to be negative for mycoplasma, and low passage cultures were applied in this experiment.

## Quantitative polymerase chain reaction (q-PCR)

Total RNA from the HNSCC cells was isolated using TRIzol reagent (Invitrogen, San Diego, CA, USA). cDNA was synthesized from 2 μg of total RNA using the PrimeScript RT Reagent kit (TaKaRa, Shiga, Japan). The ChamQ Universal SYBR qPCR Master Mix reaction system (Vazyme, Nanjing, China) was used for q-PCR with specific primers. GAPDH served as an internal control. The relative mRNA expression levels were calculated

**Table 1  The primers applied in the quantitative polymerase chain reaction of this study.**

| Primers | Species | Forward 5′–3′ | Reverse 5′–3′ |
|---------|---------|---------------|---------------|
| CASP1 | Human | GAAAAGCCATGGCCGACAAG | GCTGTCAGAGGTCTTGTGCT |
| CASP3 | Human | CATGGAAGCGAATCAATGGACT | CTGTACCAGACCGAGATGTCA |
| GSDMD | Human | GTGTGTCAACCTGTCTATCAAGG | CATGGCATCGTAGAAGTGGAAG |
| GSDME | Human | ACATGCAGGTCGAGGAGAAGT | TCAATGACACCGTAGGCAATG |
| GAPDH | Human | GAGTCAACGGATTTGGTCGT | TTGATTTTGGAGGGATCTCG |

**Notes.**

Abbreviations: CASP1, Caspase-1; CASP3, Caspase-3; GSDMD, Gasdermin D; GSDME, Gasdermin E; GAPDH, Glyceraldehyde-3-phosphate dehydrogenase.

using the 2-ΔΔCt method. The specific primers used in this study were provided by Sangon Biotech (Shanghai, China) and are listed in Table 1.

### Chemosensitivity assay

The CCK8 assay was employed to determine the impact of cisplatin (MedChemExpress, Monmouth Junction, NJ, USA) and 5-fluorouracil (5-fluorouracil; MedChemExpress, Monmouth Junction, NY, USA) on the viability of HNSCC cells. Cell lines HSC-3, CAL-27, UM1, SCC9, H103, H357, H376, H400 and SCC25 were seeded in 96-well plates at a density of $2 \times 10^3$ cells/well, while HSC-4, HN4, HN12, HN30, and H314 were seeded at $3 \times 10^3$ cells/well. The cells were then treated with varying concentrations of cisplatin or 5-Fluorouracil, followed by incubation for 72 h. Cell viability was subsequently assessed by adding a 10% CCK8 solution (Yeasen Biotechnology, Shanghai, China) to each well, incubating for an additional 2 h, and measuring the absorbance at a wavelength of 450nm using Varioskan LUX (Thermo Scientific, Waltham, MA, USA). The half-maximal inhibitory concentration (IC50) values were then calculated based on the relative survival curves.

### Statistical analysis

Differences in continuous variables between the HNSCC and normal tissues were evaluated using the Student's $t$-test and Wilcoxon test. Survival curves were compared using the log-rank test in the Kaplan–Meier analysis. Cox regression analysis was utilized to calculate the hazard ratio (HR) and 95% confidence interval (CI) of pyroptosis-related genes and clinical parameters. All statistical analyses were conducted using R software (version 4.1.1; *R Core Team, 2020*) and GraphPad Prism 9.0.

## RESULTS

### Individual pyroptosis-related gene expression is associated with poor prognosis of HNSCC

The study design is illustrated in Fig. 1. The expression levels of four pyroptosis-related genes were analyzed in 497 HNSCC and 44 normal tissue samples obtained from the TCGA database. The expression of the four genes (CASP1, CASP3, GSDMD, and GSDME) was found to be significantly elevated in HNSCC tissues (Fig. 2A). Kaplan–Meier (KM) survival analysis of the corresponding genes suggested that higher expression levels of these genes

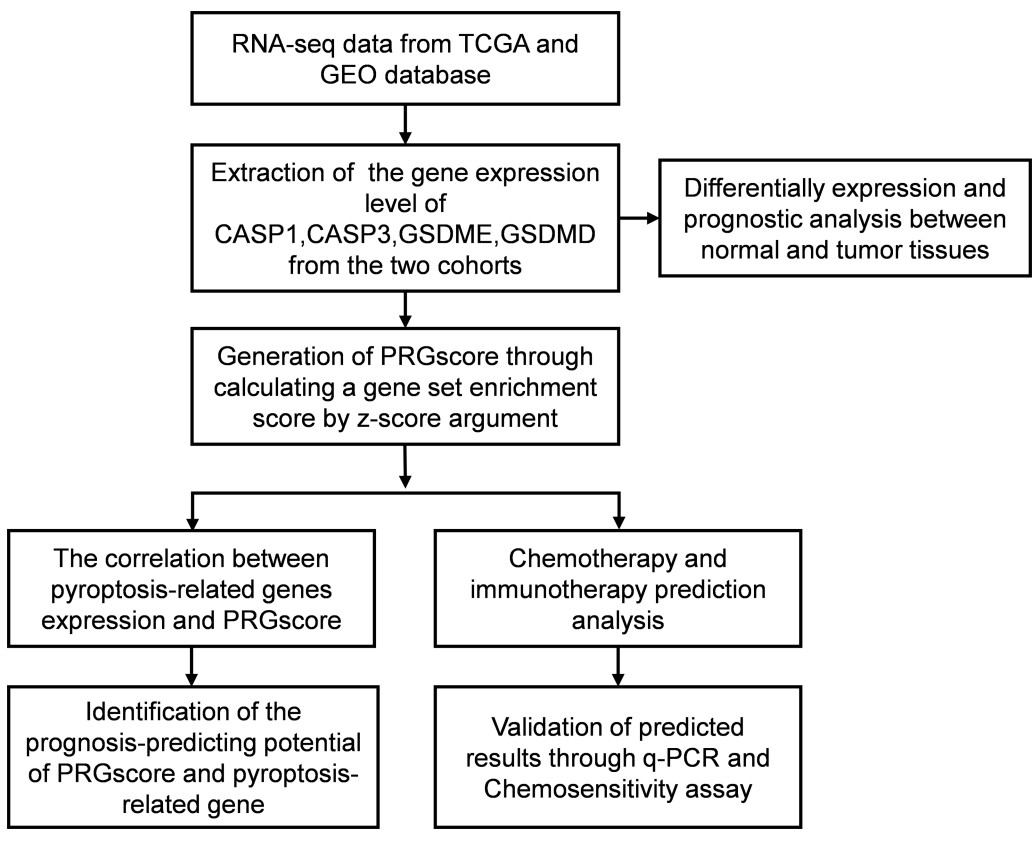

**Figure 1   Flow diagram regarding design of the study.** Abbreviations: TCGA, The Cancer Genome Atlas; GEO, The Gene Expression Omnibus; CASP1, Caspase-1; CASP3, Caspase-3; GSDMD, Gasdermin D; GSDME, Gasdermin E; PRGscore, Pyroptosis-Related Gene Score; q-PCR, Quantitative polymerase chain reaction.

were associated with worse prognosis (CASP3, $P = 0.006$; GSDME, $P = 0.009$, Fig. 2B). Moreover, independent prognostic analysis revealed that the odds ratio of these four genes were all greater than 1.00 (GSDME, $P = 0.019$, Fig. 2C). These findings suggest that the elevated expression of these four pyroptosis-related genes may be associated with worse prognosis of HNSCC.

## High PRGsocre may predict unfavorable prognosis of HNSCC

A scoring system, the pyroptosis-related gene score, was created using the GSVA method with the z-score argument to quantify the overall level of pyroptosis. Expression levels of each gene is positively correlated with the PRGscore in both the TCGA and GEO cohorts (correlation coefficient > 0.3 and $P < 0.05$, Figs. 3A and S1A). Thus, we define this PRGscore as a signature to characterize pyroptosis activity. Although no significant correlation was found between the PRGscore and clinicopathological characteristics ($P > 0.05$, Table S1). Using the "surv_cutpoint" function of the "survminer" R package, samples were categorized into high and low PRGscore groups (Figs. 3B and S1B). KM survival analysis revealed that patients with a higher PRGscore had significantly worse

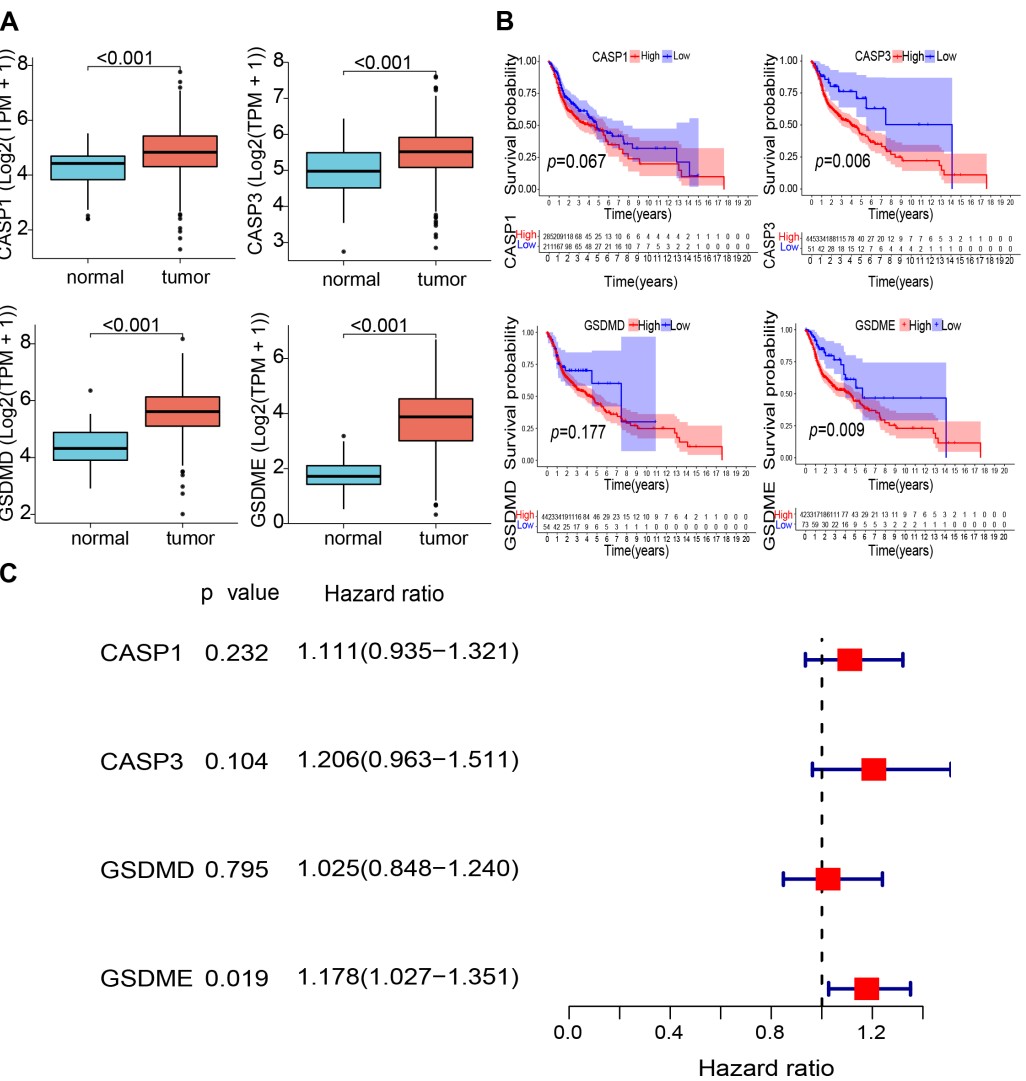

**Figure 2** **Expression and prognostic value of 4 pyroptosis-related genes in HNSCC patients.** (A) Comparison of expression of four pyroptosis-related genes between the tumor and normal tissues. (B) Kaplan–Meier curves of low and high expression subgroups. (C) Univariate Cox regression analysis of the four prognostic-related genes in HNSCC samples. Abbreviations: CASP1, caspase-1; CASP3, caspase-3; GSDMD, gasdermin D; GSDME, gasdermin E; TPM, transcripts per million value; HNSCC, head and neck squamous cell carcinoma.

prognosis (Fig. 3C, $P = 0.014$), which was verified in GEO cohort (Fig. S1C, $P = 0.017$). Multivariate Cox regression analysis demonstrated that PRGscore, as a categorical variable, was an independent predictor of inferior prognosis in both the TCGA and GEO cohorts (Fig. 3D, hazard ratio of PRGscore $= 1.473$, $P = 0.007$; Fig. S1D, hazard ratio of PRGscore $= 1.840$, $P = 0.034$). Meanwhile, the COX proportional hazards assumption was verified, with the result indicating that each covariate has a smooth fitting curve of Schoenfeld residuals changing over time that is almost parallel to the $x$-axis with $p$ valu $e > 0.05$ (Fig. S4 and Table S2), thus confirming the applicability multivariate Cox regression analysis in this

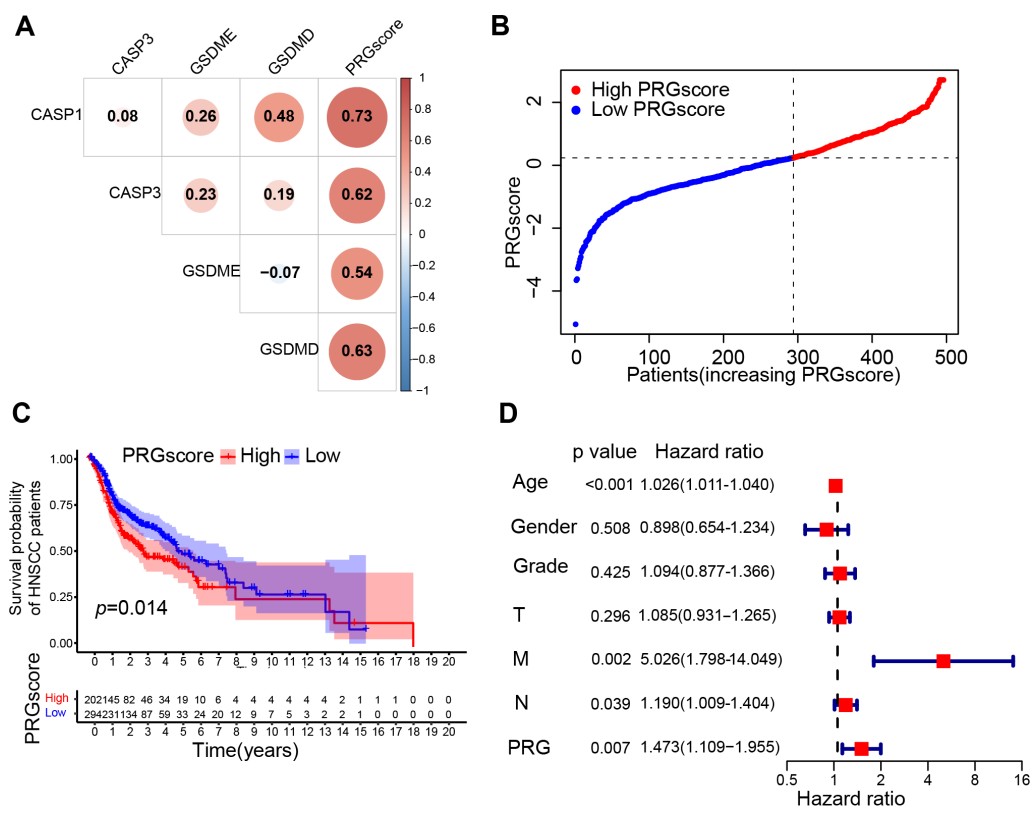

**Figure 3** **Construction of the pyroptosis-related prognostic signature (TCGA cohort).** (A) Pearson correlation analysis between 4 pyroptosis-related genes and PRGscore. (B) PRGscore of HNSCC samples premised on the cutpoint (the left of the dotted line signifies the low PRGscore group while the right side signifies the high PRGscore group). (C) Kaplan–Meier survival analysis of low and high PRGscore sub-groups. (D) Multivariate Cox regression analysis of the PRGscore and clinicopathological parameters in HNSCC samples. Abbreviations: TCGA, The Cancer Genome Atlas; CASP1, caspase-1; CASP3, caspase-3; GSDMD, gasdermin D; GSDME, gasdermin E; PRGscore, pyroptosis-related gene score; HNSCC, head and neck squamous cell carcinoma.

context. Collectively, these results suggest that a higher PRGscore is associated with worse prognosis in HNSCC.

## High PRGscore serves as a protective factor in the cohort of patients receiving chemotherapy

To explore the impact of pyroptosis activity on response to chemotherapy treatment, we investigated whether high pyroptosis levels were a protective factor for prognosis in the chemotherapy cohort and the non-chemotherapy cohort within the TCGA HNSCC dataset. The KM survival analysis and multivariate Cox regression analysis indicated that a higher PRGscore was associated with worse prognosis in the patients not implementing chemotherapy ($P = 0.018$, Log-rank test, Fig. 4A; hazard ratio of PRGscore = 1.523, $P = 0.006$, Fig. 4B). However, patients within the higher PRGscore subgroup exhibited better prognosis in the chemotherapy cohort ($P = 0.05$, Log-rank test, Fig. 4C; hazard ratio of PRGscore = 0.175, $P = 0.057$, Fig. 4D). Given the observed association between higher

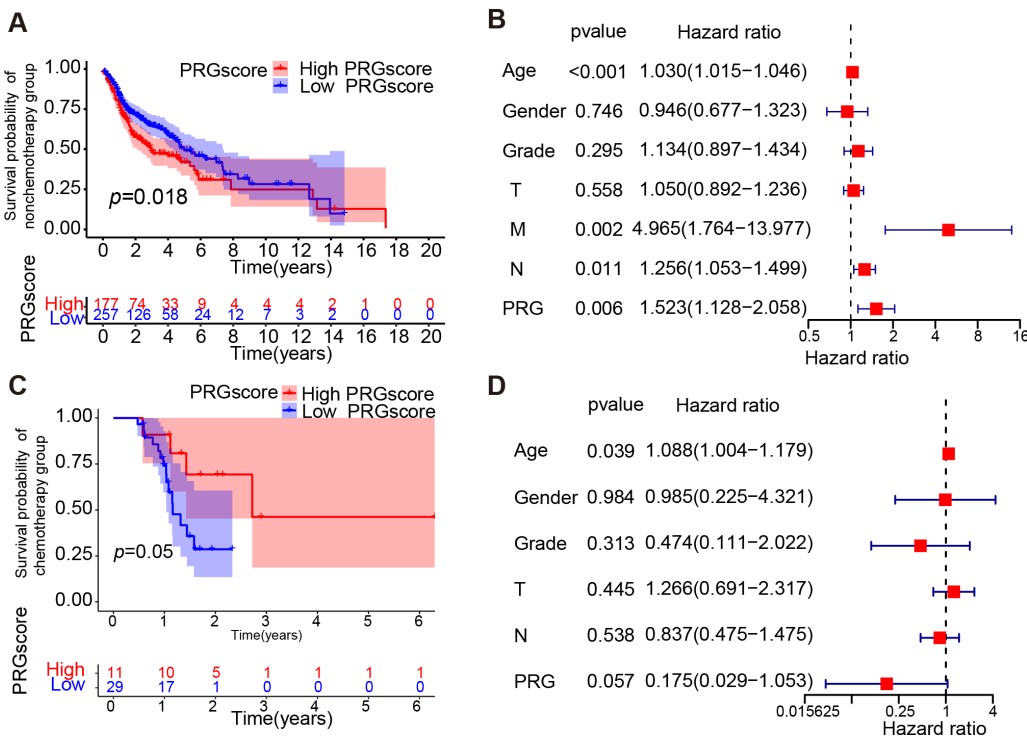

**Figure 4 Correlation between PRGscore and chemotherapy.** (A) Kaplan–Meier curves of low and high PRGscore groups in HNSCC patients not receiving chemotherapy. (B) Multivariate Cox regression analysis of the PRGscore (as a categorical variable) in HNSCC patients not receiving chemotherapy. (C) Kaplan–Meier curves of low and high PRGscore groups in HNSCC patients adopting chemotherapy. (D) Multivariate Cox regression analysis of the PRGscore (as a categorical variable) in HNSCC patients adopting chemotherapy. Abbreviations: PRGscore, pyroptosis-related gene score; HNSCC, head and neck squamous cell carcinoma.

levels of pyroptosis and better prognosis in the chemotherapy HNSCC patients, further investigation into the relationship between PRGscore and chemotherapeutic sensitivity is warranted.

## High PRGscore may suggest better chemotherapeutic sensitivity in HNSCC

To initially investigate the potential association between PRGscore and chemotherapeutic sensitivity, the half-maximal inhibitory concentration (IC50) of commonly used chemotherapeutic agents for HNSCC was estimated using the pRRophetic algorithm. As hypothesized, our analysis revealed a negative correlation between PRGscore and the IC50 values of chemotherapeutic agents such as docetaxel, gemcitabine, paclitaxel, methotrexate, cisplatin, and 5-fluorouracil in HNSCC. This finding implies that patients with a high PRGscore may show increased sensitivity to chemotherapy (Figs. 5A–5B, $P < 0.05$).

To further verify the presumed relationship between overall pyroptosis levels and chemotherapy sensitivity, the IC50 values of cisplatin (Fig. 6) and 5-fluorouracil (Fig. 7)

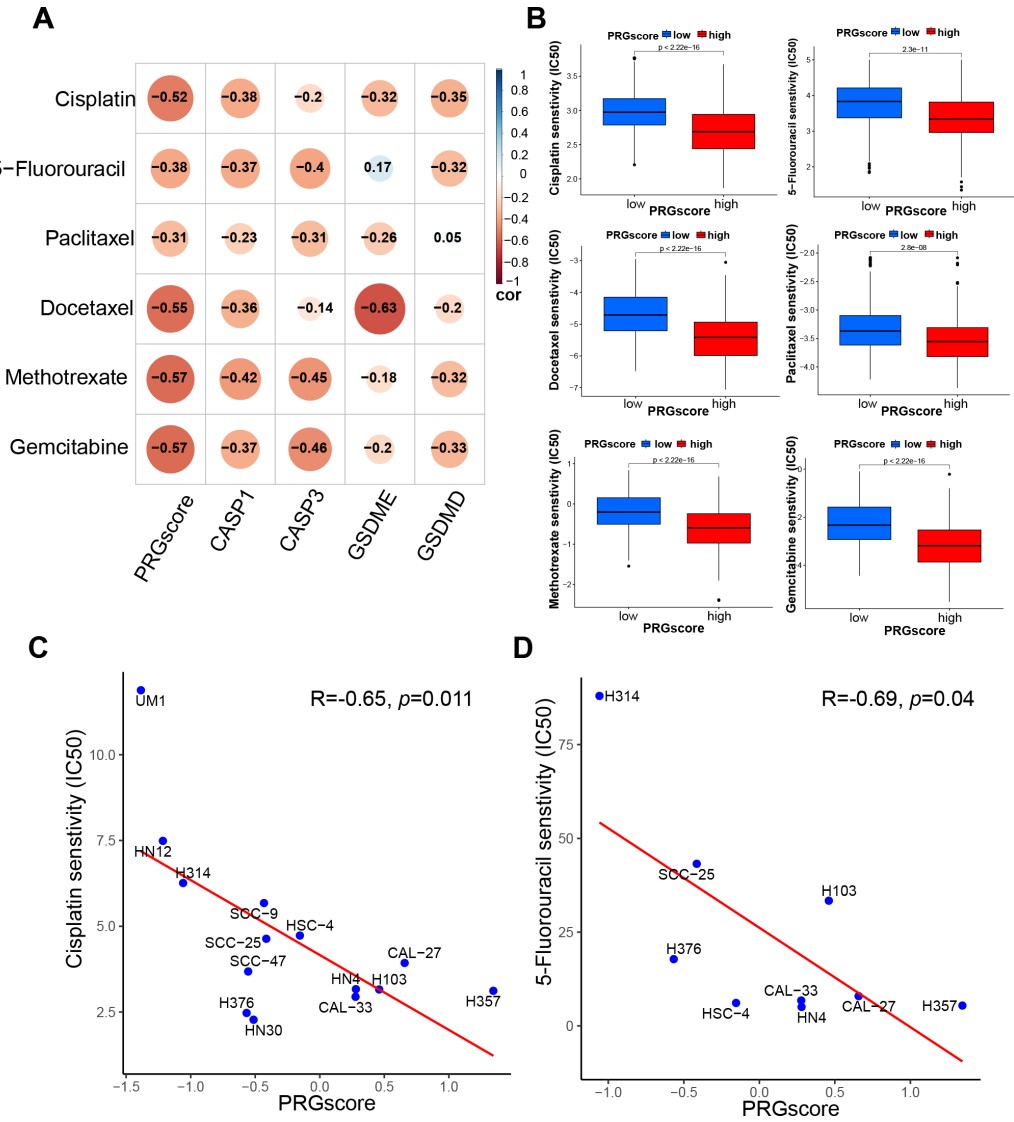

**Figure 5** **Correlation between PRGscore and chemotherapy sensitivity through pRRophetic R package and verified in HNSCC cell lines.** (A) Chemotherapy sensitivity prediction analysis through pRRophetic R package. (B) Drug sensitivity analysis of chemotherapy for HNSCC patients between high- and low-PRGscore cohorts. (C) Pearson correlation analysis between the PRGscore and the IC50 of cisplatin in HNSCC cell lines. (D) Pearson correlation analysis between the PRGscore and the IC50 of 5-fluorouracil in HNSCC cell lines. Abbreviations: CASP1, caspase-1; CASP3, caspase-3; GSDMD, gasdermin D; GS-DME, gasdermin E; PRGscore, pyroptosis-related gene score; HNSCC, head and neck squamous cell carcinoma.

were evaluated in multiple HNSCC cell lines, along with the expression levels of the four pyroptosis-related genes (Fig. S2). Utilizing the same analytical method, the PRGscore of each cell line was calculated. As expected, Pearson correlation analysis revealed significantly negative relationship between IC50 of these HNSCC cells for cisplatin or 5-fluorouracil and pyroptosis activity of these cell lines, indicating that HNSCCs with higher PRGscore are

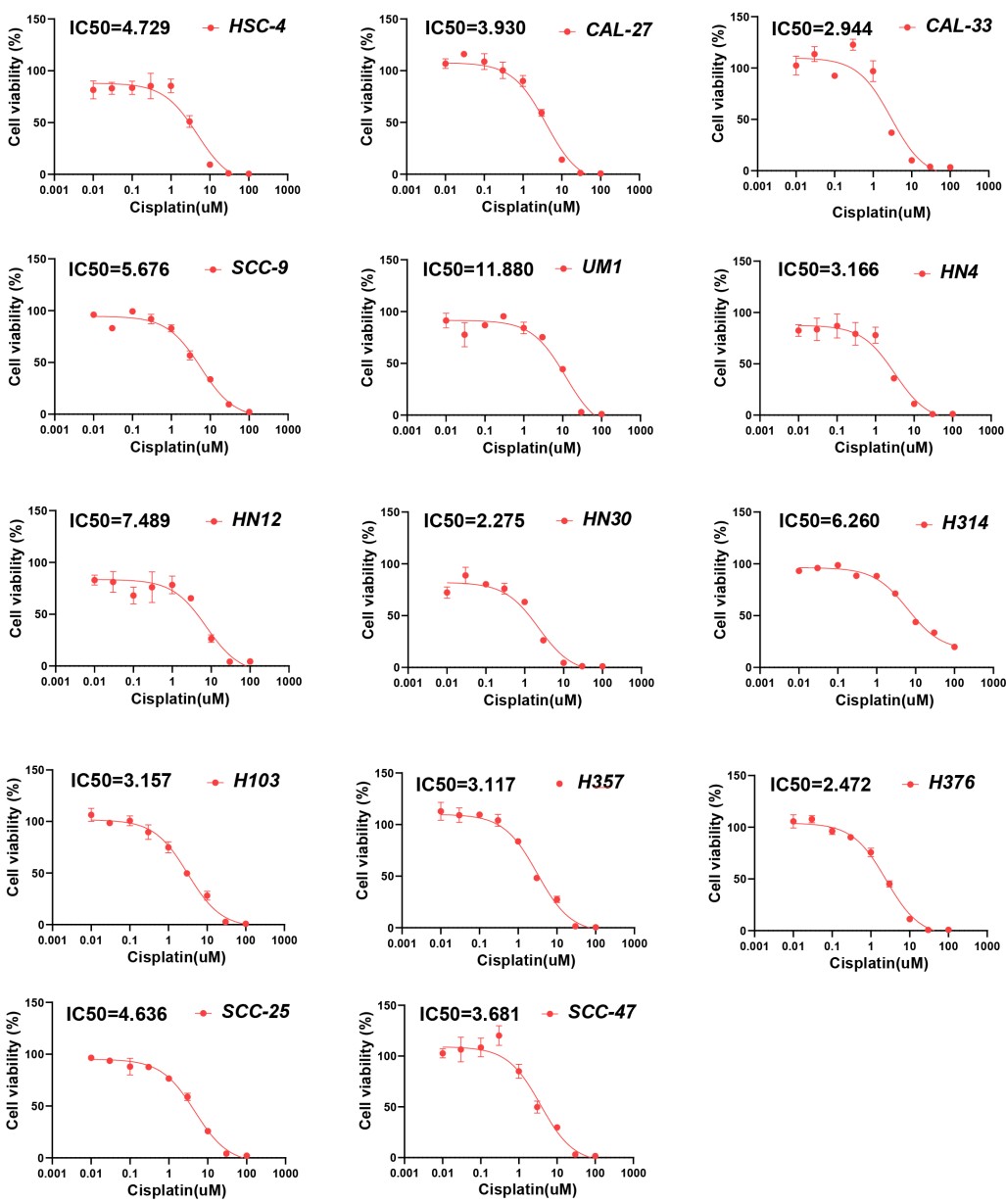

**Figure 6** **Chemosensitivity assay of cisplatin for multiple HNSCC cell lines.** Determination of IC50 values of cisplatin for HNSCC cell lines using chemosensitivity assay. Abbreviations: IC50, the half-maximum inhibitory concentration; HNSCC, head and neck squamous cell carcinoma.

more sensitive to chemotherapy (Fig. 5C, R = −0.65, P = 0.011 for cisplatin and Fig. 5D, R = −0.69, P = 0.04 for 5-fluorouracil).

## High PRGsocre may be associated with better efficacy of immunotherapy

Furthermore, we conducted correlation analysis of gene expression and immunotherapy to investigate the potential role of PRGscore in predicting patients' response to

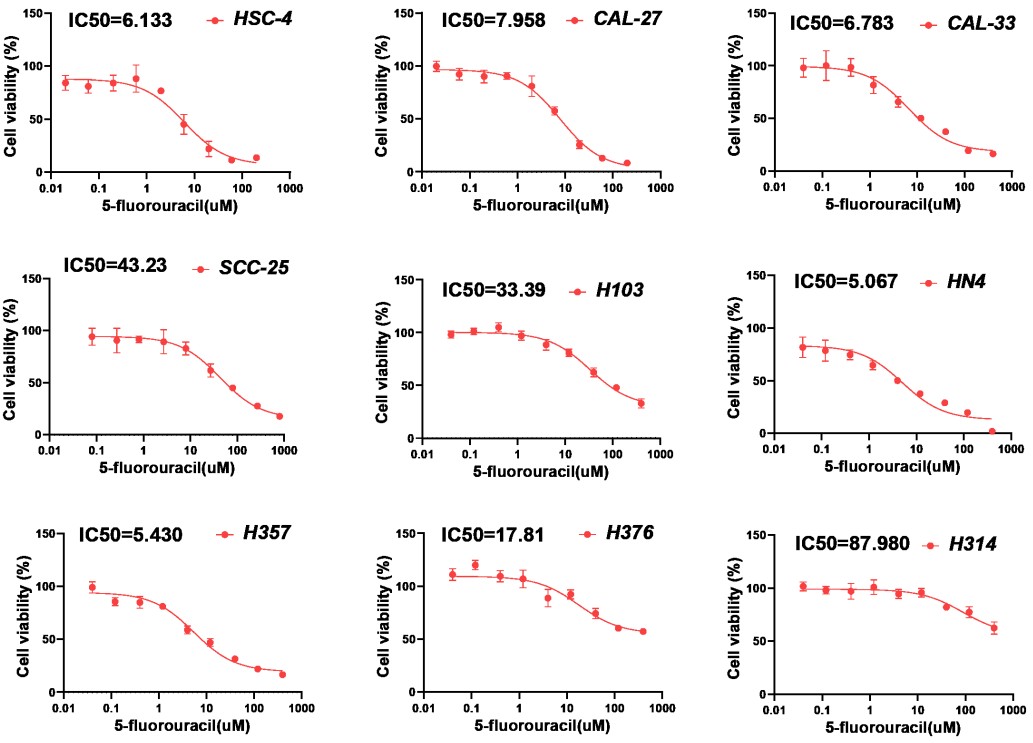

**Figure 7 Chemosensitivity assay of 5-fluorouracil for multiple HNSCC cell lines.** Determination of IC50 values of 5-fluorouracil for HNSCC cell lines by applying chemosensitivity assay. Abbreviations: IC50, the half-maximum inhibitory concentration; HNSCC, head and neck squamous cell carcinoma.

immunotherapy. The results of the immune checkpoint analysis demonstrated a positive correlation between PRGscore and several key immune checkpoints, including CD274, CTLA4, PDCD1, IDO1, PDCD1LG2, and HAVCR2. Notably, the high PRGscore group exhibited a tendency toward higher expression levels of these immune checkpoints significantly, as illustrated in Figs. S3A and S3B. Immunophenoscore (IPS) data for patients diagnosed with HNSCC were obtained from the Cancer Immuneome Database (TCIA). The analysis of IPS data revealed that individuals with elevated PRGscore or increased pyroptosis gene expression may benefit more from immunotherapy targeting PD1, CTLA4, or the combination of CTLA4 and PD1, as depicted in Fig. S3C to S3G. These suggested that individuals with higher PRGscore may benefit from specific immunotherapies.

## DISCUSSION

The dual impact of pyroptosis on tumorigenesis is well-acknowledged. On the one hand, it constrains tumor growth through programmed cell death, and on the other hand, it fosters a microenvironment conducive to tumor cell survival due to the inflammatory response it triggers (*Wang et al., 2019*; *Xia et al., 2019*). In the context of the progression of head and neck squamous cell carcinoma (HNSCC), the dual impact of pyroptosis in either promoting or inhibiting carcinogenesis remains incompletely understood. Consequently, we have been motivated to undertake an analysis of pyroptotic activity

and its correlation with the prognosis among patients within the TCGA HNSCC cohort. Recently, pyroptotic cell death has been suggested as a novel molecular mechanism underpinning the effectiveness of various anticancer agents, including chemotherapy (*Wang et al., 2017*). Previous researchers have documented that chemotherapy has the potential to shift cell death from apoptosis to pyroptosis through caspase-3-mediated GSDME cleavage, particularly in tumor cells exhibiting elevated GSDME expression (*Wang et al., 2017*; *Rogers et al., 2017*). At the same time, many studies have proven that high levels of pyroptosis genes in tumor cells contribute to anti-tumor drug sensitivity (*Lage et al., 2001*; *Lu et al., 2018*; *Yu et al., 2019*; *Ying et al., 2019*). Additionally, it has been reported that both paclitaxel and cisplatin can activate the caspase-3/GSDME pathway, shifting the cell death process from apoptosis to pyroptosis, thus suppressing lung cancer cell proliferation (*Zhang et al., 2019*). Other studies have indicated that 5-fluorouracil can trigger caspase-3 activation and GSDME cleavage in gastric cancer cells, thereby converting caspase-3-dependent apoptosis into pyroptosis (*Wang et al., 2018*). Moreover, evidence shows that cisplatin can enhance caspase-3 function, contributing to the pyroptosis of esophageal cancer cells, and simultaneously augmenting DNA damage (*Wu et al., 2019*). These findings advance our understanding of the beneficial effects of caspase-3-dependent pyroptosis induction in cancer cells *via* chemotherapeutic drugs. However, the association of pyroptosis-related genes with chemotherapeutic sensitivity in HNSCC have not been extensively investigated. HNSCC patients tend to develop chemotherapeutic resistance and suffer heavy side effects induced by the treatment; thus, it is significant and highly required to identify the patients who might be more susceptible to the therapy based on the pyroptosis activity, which has not been comprehensively investigated in the previous studies.

In the present study, we have developed a pyroptosis-related gene score (PRGscore) using GSVA with the z-score argument to quantify the overall pyroptosis level. After establishing a positive correlation between the expression of each pyroptosis-related gene and the PRGscore, we found that higher PRGscore were associated with worse prognosis of HNSCC. Our study, similar to these previous studies (*Lu et al., 2022*; *Shen et al., 2021*; *Zhu et al., 2021*; *Li et al., 2022*; *Liu et al., 2019*), focuses on identifying new biomarkers and predictive models to improve the prognosis and treatment response of cancer patients through machine learning. Prior studies have generally utilized the least absolute shrinkage and selection operator (LASSO) to construct pyroptosis-related molecular features for predicting the survival and prognosis of HNSCC patients (*Lu et al., 2022*; *Shen et al., 2021*; *Zhu et al., 2021*; *Li et al., 2022*), which is markedly differed from our study design in two key aspects. First, as most previous studies have identified several key genes from 33 pyroptosis genes using a LASSO regression model (*Breiman, 1995*; *Tibshirani, 1996*), our study aims to construct a new prognostic prediction signature based on 4 genes reported to implicate in the pyroptosis of HNSCC. Second, those previous studies constructed the gene signature primarily based on the correlation between gene expression and patients' survival, while neglecting its relationship with pyroptotic activity; in contrast, the PRGscore we computed using the GSVA package utilizing the z-score gene set enrichment method is appropriate for presenting the real pyroptotic level (*Lee et al., 2008*), which is for the sake that the PRGscore has been validated to be positively related with expression of each of the 4 genes

reported to mediate pyroptosis of HNSCC. Thus, the PRGscore algorithm allows us not only to predict the relationship between overall pyroptosis level and HNSCC prognosis, but also project the chemotherapeutic sensitivity.

Our analysis revealed that higher pyroptotic activity was a negative prognosticator for HNSCC, with individual analyses of GSDME or CASP3 also demonstrating similar associations with the prognosis; considering that non-significant correlation was revealed between expression of CASP1 or GSDMD with prognosis, and the prognostic prediction effect of the 4-gene based signature was slightly inferior to that of GSDME or CASP3, we have evaluated the joint prediction performance of GSDME and CASP3 in HNSCC prognosis of chemotherapy or non-chemotherapy cohorts, as well as in the chemotherapy sensitivity of HNSCC (Figs. S5–S7). The results suggest that the four-gene based model has better predicting performance; therefore, the four-gene model was selected for further analysis. A study affirmed that high levels of pyroptosis were associated with inferior prognosis in kidney renal clear cell carcinoma and thymoma, even though the adverse prognostic value in HNSCC was not statistically significant (*Lou et al., 2022*). The results align with our findings and propose a potential mechanism whereby high pyroptosis levels promote malignant progression and poor prognosis. Specific molecular mechanisms accounting for the pyroptosis-related tumor development include the establishment of a pro-tumorigenic microenvironment due to the release of pro-inflammatory cytokines during pyroptosis and the triggering of tumor cell immune evasion by damage-associated molecular patterns (DAMPs) (*Jia et al., 2023*). However, a more in-depth exploration of these molecular mechanisms is required.

Interestingly, after dividing the patients into chemotherapy and non-chemotherapy groups, we found that patients with high pyroptosis levels may derive more benefit from chemotherapy. Previous studies have linked the endogenous expression of GSDME to chemotherapy responsiveness. For example, reduced GSDME expression has been shown to compromise the effectiveness of anti-tumor treatments, while increased GSDME expression could potentially enhance the efficacy of these therapies (*Wang et al., 2017*; *Zhang et al., 2019*; *Jia et al., 2023*; *Chen et al., 2022*). Our initial results suggested a strong relationship between overall pyroptotic level and chemotherapeutic response, prompting further studies to investigate the association between PRGscore and chemotherapeutic sensitivity.

To further investigate this, we first conducted a bioinformatics study. The "pRRophetic" package was used to calculate the IC50 of conventional chemotherapy drugs for HNSCC. Previous study showed that HNSCC patients with low pyroptotic scores tended to be more resistant to most chemotherapeutic agents, including paclitaxel, docetaxel, and cisplatin in the same way (*Deng et al., 2022*), which supports our above analysis. Subsequently, a drug sensitivity prediction tool was used to estimate the correlation between PRGscore and the IC50 of the chemotherapeutic drugs. As expected, the pooled pyroptotic activity was statistically positively correlated with chemotherapeutic drug sensitivity. To further verify these findings, chemosensitivity assays were conducted on 14 head and neck squamous cell lines. As expected, these results were consistent with the drug sensitivity prediction results obtained from the TCGA bioinformatics analysis.

Besides, another analysis results of us suggest that higher PRGscore or pyroptosis gene expression level predict better immunotherapy efficacy, which is consistent with previous relevant research findings (*Zhang et al., 2022*; *Wang, Zhang & Sun, 2022*). In the future, the potential of utilizing our PRGscore in predicting efficacy of immunotherapy should be validated in large clinical cohorts.

In all, our study uncovers that although high PRGscore is an indicator of poor prognosis, patients with higher PRGscores might be more suitable candidates for receiving chemotherapy, possibly the same for immunotherapy, in the clinical decision-making processes, as optimal efficacy tend to be reflected in this special population. However, more research is needed to validate our findings.

## CONCLUSION

In summary, our study constructs a novel predictive model, the PRGscore, which is based on the expression of four pyroptosis-related genes and is specifically designed for prognosticating and gauging chemotherapy sensitivity in HNSCC. Our findings, substantiated through bioinformatic analysis and experimental validation, indicate that patients with higher PRGscore, despite facing worse prognosis, could potentially obtain better benefits from chemotherapy. Consequently, we propose that the PRGscore could serve as a valuable instrument in guiding clinical decisions, particularly in terms of administering chemotherapy for HNSCC. This study, however, underscores the need for further validation of our model in larger and diverse patient cohorts to confirm its clinical utility.

### Funding

This work is supported by the National Natural Science Foundation of China (82272899, 81902782, 82203180, 51733005, and 52173287), the Research Funding from West China School/Hospital of Stomatology Sichuan University (No.RCDWJS2022-16), the Postdoctoral Research Funding of Sichuan University (2022SCU12132), the Research and Develop Program of West China Hospital of Stomatology of Sichuan University (No. RD-02-202204), the Key Research Program of Sichuan Provincial Science and Technology Agency (2023YFS0127), the Clinical and Translational Medicine Research Foundation of Chinese Academy of Medical Sciences (2022-I2M-C&T-B-111), and the Research and Develop Program, West China Hospital of Stomatology Sichuan University (RD-03-202307). The funders had no role in study design, data collection and analysis, decision to publish, or preparation of the manuscript.

### Grant Disclosures

The following grant information was disclosed by the authors:
National Natural Science Foundation of China: 82272899, 81902782, 82203180, 51733005, 52173287.

Research Funding from West China School/Hospital of Stomatology Sichuan University: No. RCDWJS2022-16.
Postdoctoral Research Funding of Sichuan University: 2022SCU12132.
Research and Develop Program of West China Hospital of Stomatology of Sichuan University: No. RD-02-202204.
Key Research Program of Sichuan Provincial Science and Technology Agency: 2023YFS0127.
Clinical and Translational Medicine Research Foundation of Chinese Academy of Medical Sciences: 2022-I2M-C&T-B-111.
Research and Develop Program, West China Hospital of Stomatology Sichuan University: RD-03-202307.

## Competing Interests

The authors declare there are no competing interests.

## Author Contributions

- Peiyang Yuan performed the experiments, prepared figures and/or tables, authored or reviewed drafts of the article, and approved the final draft.
- Sixin Jiang performed the experiments, prepared figures and/or tables, authored or reviewed drafts of the article, and approved the final draft.
- Qiuhao Wang performed the experiments, prepared figures and/or tables, authored or reviewed drafts of the article, and approved the final draft.
- Yuqi Wu performed the experiments, prepared figures and/or tables, authored or reviewed drafts of the article, and approved the final draft.
- Yuchen Jiang analyzed the data, prepared figures and/or tables, authored or reviewed drafts of the article, and approved the final draft.
- Hao Xu analyzed the data, prepared figures and/or tables, authored or reviewed drafts of the article, and approved the final draft.
- Lu Jiang conceived and designed the experiments, authored or reviewed drafts of the article, and approved the final draft.
- Xiaobo Luo conceived and designed the experiments, authored or reviewed drafts of the article, and approved the final draft.

## Data Availability

The relative mRNA expression and chemosensitivity assay are available in the Supplemental Files.

The TCGA-HNSCC dataset is available at: https://portal.gdc.cancer.gov/projects/TCGA-HNSC.

The GEO dataset is available at NCBI: GSE41613.

The data required for drug sensitivity analysis was downloaded from the software package "pRRophetic" (*Geeleher, Cox & Huang, 2014*).

## Supplemental Information

Supplemental information for this article can be found online at http://dx.doi.org/10.7717/peerj.17296#supplemental-information.

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
