# Peer review of "Prognostic and chemotherapeutic implications of a novel four-gene pyroptosis model in head and neck squamous cell carcinoma"

_PeerJ, doi:10.7717/peerj.17296_

## Round 0.1 · original submission · Major Revisions

Your manuscript has been reviewed by two experts in the field, each of whom have raised a number of different issues that you will need to address by revising the manuscript. Please note that Reviewer 2 has provided their comments as a separate file, which I have attached here.

I hope you find the reviewer comments constructive and helpful.

Reviewer 1 ·

Basic reporting

The study brings a significant contribution towards understanding pyroptosis's role in HNSCC, potentially influencing chemotherapy approaches. The introduced multi-gene model is quite fascinating and deserves more in-depth study. Yet, addressing the raised concerns about statistical depth, clarity in visualization, logic in research, and experiment specifics would greatly bolster the manuscript's impact and trustworthiness. I reckon it could be ready for publication post the advised minor tweaks.

Experimental design

1. A deeper dive into the survival analysis details, especially around the Cox proportional hazards model's assumptions and their verification, would solidify the statistical findings' foundation.
2. Adding confidence intervals to the Kaplan-Meier survival plots can offer a clearer picture of survival probabilities over time, enhancing data interpretation.
3. Considering your innovative four-gene pyroptosis model for HNSCC, the study 'A four-pseudogene classifier identified by machine learning serves as a novel prognostic marker for survival of osteosarcoma' could be very pertinent. Their method of crafting a prognostic model via machine learning and pseudogene analysis might offer insightful parallels and insights. Mentioning this research could enrich your model's discussion, spotlighting its wider impact and innovative aspect.
4. The paper suggests a new four-gene pyroptosis model but lacks a full discussion on why these genes were picked over others. Delving into the biological significance and selection logic for these genes would enhance the paper's logical progression and scientific grounding.

Validity of the findings

More precise details on cell culture conditions, like specific passage numbers for in vitro tests, are crucial for reproducibility and contextual understanding of the experiments' outcomes.

Reviewer 2 ·

Basic reporting

no comment

Experimental design

The research question is well defined but not well studied using the current experimental design. The rationality in gene selection should be better explained. Otherwise, other bioinformatics methods should be applied to identify best performing models for predicting the response to chemotherapy.

Validity of the findings

Some results should be described and interpreted properly.

Annotated reviews are not available for download in order to protect the identity of reviewers who chose to remain anonymous.

---

## Round 0.2 · accepted · Accept

Many thanks for addressing the previous comments from the two reviewers. They are both content with your responses.

Reviewer 1 ·

Basic reporting

Upon careful consideration of the revised manuscript, I am impressed by the substantial improvements made, effectively elevating its quality to fulfill our publication criteria. The authors' dedication to addressing the feedback received previously is praiseworthy, resulting in a manuscript that convincingly justifies its acceptance. I wholeheartedly endorse its publication and am enthusiastic about its potential impact on our field of research.

Experimental design

none

Validity of the findings

none

Reviewer 2 ·

Basic reporting

no comment

Experimental design

no comment

Validity of the findings

no comment

Additional comments

I have no further comments as the authors have properly addressed my previous comments in the revised version of the manuscript.